# Drought Monitoring Using Landsat Derived Indices and Google Earth Engine Platform: A Case Study from Al-Lith Watershed, Kingdom of Saudi Arabia

Nuaman Ejaz [1], Jarbou Bahrawi [1], Khalid Mohammed Alghamdi [1], Khalil Ur Rahman [2] and Songhao Shang [2,*]

[1] Department of Hydrology and Water Resources Management, Faculty of Meteorology, Environment & Arid Land Agriculture, King Abdulaziz University, Jeddah 21589, Saudi Arabia
[2] State Key Laboratory of Hydro Science and Engineering, Department of Hydraulic Engineering, Tsinghua University, Beijing 100084, China
* Correspondence: shangsh@tsinghua.edu.cn; Tel.: +86-10-6279-6674

**Abstract:** Precise assessment of drought and its impact on the natural ecosystem is an arduous task in regions with limited climatic observations due to sparsely distributed in situ stations, especially in the hyper-arid region of Kingdom of Saudi Arabia (KSA). Therefore, this study investigates the application of remote sensing techniques to monitor drought and compare the remote sensing-retrieved drought indices (RSDIs) with the standardized meteorological drought index (Standardized Precipitation Evapotranspiration Index, SPEI) during 2001–2020. The computed RSDIs include Vegetation Condition Index (VCI), Temperature Condition Index (TCI), and Vegetation Health Index (VHI), which are derived using multi-temporal Landsat 7 ETM+, Landsat 8 OLI/TIRS satellites, and the Google Earth Engine (GEE) platform. Pearson correlation coefficient (CC) is used to find the extent of agreement between the SPEI and RSDIs. The comparison showed CC values of 0.74, 0.67, 0.57, and 0.47 observed for VHI/SPEI-12, VHI/SPEI-6, VHI/SPEI-3, and VHI/SPEI-1, respectively. Comparatively low agreement was observed between TCI and SPEI with CC values of 0.60, 0.61, 0.42, and 0.37 observed for TCI/SPEI-12, TCI/SPEI-6, TCI/SPEI-3, and TCI/SPEI-1. A lower correlation with CC values of 0.53, 0.45, 0.33 and 0.24 was observed for VCI/SPEI-12, VCI/SPEI-6, VCI/SPEI-3, and VCI/SPEI-1, respectively. Overall, the results suggest that VHI and SPEI are better correlated drought indices and are suitable for drought monitoring in the data-scarce hyper-arid regions. This research will help to improve our understanding of the relationships between meteorological and remote sensing drought indices.

**Keywords:** drought assessment; meteorological drought; remote sensing drought indices; standardized drought indices; Landsat; Google Earth Engine

## 1. Introduction

It is challenging to precisely monitor and evaluate the onset, intensity, frequency, persistence, and propagation of drought because of its complex nature, especially in hyper-arid regions characterized by data scarcity [1,2]. Drought is a frequently occurring hydrometeorological phenomenon [3], which is the direct cause of drought disasters and the second most detrimental natural hazard that causes social and economic instability after floods [4]. Drought events are categorized into four categories based on the affected sectors [5] including meteorological, hydrological, agricultural, and socio-economic droughts. Meteorological drought is characterized by an extended period of below-average precipitation (i.e., precipitation deficit) in relation to the region's average precipitation. In contrast, agricultural drought could be described as a drought resulting from soil moisture content below the level required for plant growth and development [6–9]. Hydrological drought refers to a decrease in the quantity of water both on surface and groundwater resources due

to insufficient precipitation for an extended period [10]. Socio-economic drought focuses on the consequences of drought on water resources, agriculture, and industries [11,12].

Several approaches have been developed over the last few decades to monitor and statistically describe droughts, including the development of both standardized and unstandardized drought indices used in meteorology, hydrology, and agriculture [1–3]. In the past, drought monitoring methods were based on measurements taken at stations/gauges on the ground, such as the Palmer drought Severity Index (PDSI) [4], the Standardized Precipitation Index (SPI) [5], and the Standardized Precipitation Evapotranspiration Index (SPEI) [6]. Traditional approaches for assessing and monitoring drought depend on the in situ precipitation records, which are usually inaccurate and constrained both in time and space [7,8]. The sparse distribution of in situ weather stations per unit area and the associated uncertainties hinders the precise estimation of drought, which is most often for areas in arid and hyper-arid regions. Other natural impediments, such as mountains and dune fields, can also contribute to the said problem [9]. El Kenawy and McCabe [10] have confirmed these flaws in the meteorological network over Kingdom of Saudi Arabia (KSA). However, with the advancements in remote sensing and earth observation technologies (e.g., the launch of the National Aeronautics Space Administration (NASA) Landsat series in 1972) at the end of the 20th century, the way for drought monitoring has been changed [11]. In addition, there is an increasing curiosity and understanding regarding the climate change due to rising temperatures. This has led geospatial scientists to conclude that remote sensing must play a crucial role in delivering the data necessary to assess ecosystem conditions and monitor extreme climatic changes at both spatial and temporal scales [12–15].

Remote sensing (RS) products not only provide meteorological data but also monitor changes in the variables at the earth's surface such as the health of plants and the amount of available water, and provide a wide range of contextual data for monitoring drought [11]. RS and Geographic Information Systems (GIS) have made it easier for people to look at the world with sensors and see how it changes over time [16]. The main benefit of using RS and GIS techniques is the availability of continuous data over large areas in both space and time, which significantly contributed to the data scarcity issues as we might face in arid regions like KSA [17,18].

With the advancement of RS and GIS techniques, several remote-sensing-based drought indices are proposed and evaluated to monitor drought, including the Normalized Difference Vegetation Index (NDVI) [19], the Temperature Condition Index (TCI) [20,21], the Vegetation Condition Index (VCI) [22], and the Vegetation Health Index (VHI) [23,24]. TCI, VCI, and VHI are also characterized as vegetation indices since they describe the vegetation condition in a specific area, classify it into different drought classes, and are commonly employed as drought monitoring indices [25–27]. VCI is widely used to detect changes in vegetation from significantly worse to favorable conditions [20,28]. TCI detects vegetative stress induced by high temperatures and heavy moisture [29–31]. VHI, on the other hand, is the combination of TCI (temperature) and VCI (vegetation condition) that describes vegetation health [32,33].

Recently, the Google Earth Engine (GEE), a cloud-based geospatial data monitoring platform that calculates and presents raw and processed satellite-based datasets [33–35], is extensively used in various hydro-meteorological applications. Since its introduction in 2010, GEE capabilities have been tested in a variety of applications, including vegetation mapping and monitoring [36,37], land use/land cover change mapping [38,39], and flood mapping [40,41]. Furthermore, with a substantial volume of freely available satellite imageries and direct image processing, GEE has been proposed for time series analysis of drought in several studies [37,42,43]. Most of the researchers used the Moderate Resolution Imaging Spectroradiometer (MODIS) satellite dataset and GEE algorithms to monitor drought using the Remote Sensing-retrieved Drought Indices (RSDIs) [44–46]. Meanwhile, Pham and Tran [34] analyzed the temporal distribution of drought conditions in Vietnam using different Landsat-derived drought indices, which are calculated from Landsat-8

satellite data in the GEE platform. Benzougagh et al. [47] also used the GEE algorithm and a combination of Landsat-8 and Sentinel-2 datasets to monitor drought in Morocco. The above studies showed that Landsat-derived indices provide helpful spatial information for assessing drought conditions from the region to the country scales. The main theme of this research is to address the data scarcity issues in Lith watershed. There are four gauges in the watershed which are not sufficient to represent the variations in climate. Therefore, different remote sensing techniques should be applied to cope with data scarcity issues and thus we computed different remote sensing indices using the GEE.

Owing to the condition of data scarcity and minimal application of RS, GIS, and GEE applications in drought monitoring in KSA, this study aims: (1) to calculate the spatial extent of drought in the arid basin (Al-Lith watershed) using Landsat 7 and Landsat 8 datasets from 2001 to 2020; (2) to characterize the spatiotemporal pattern of drought conditions by SPEI, VCI, and TCI, and VHI using GEE; and (3) to compare the reliability of various drought indices (particularly SPEI with RSDIs) in drought monitoring by calculating the Pearson correlation coefficient (CC). The findings in this research will assist urban planners and environmental scientists in making decisions and implementing policies to mitigate drought in the Al-Lith watershed, particularly and other similar areas around the globe in general.

## 2. Study Area

The Al-Lith watershed is located between 20°00′N to 20°15′N longitude, and 40°10′E to 40°50′E latitude in the Makkah region of KSA, with elevation ranges from 0 to 2663 m above mean sea level (shown in Figure 1). The Al-Lith watershed has a total area of about 3262 km². The maximum temperature in Al-Lith is observed in July with an average of 41.9 °C, followed by June and August with the average temperatures of 41.3 °C and 41.2 °C, respectively [48]. A minimum of 20.0 °C temperature is observed in January.

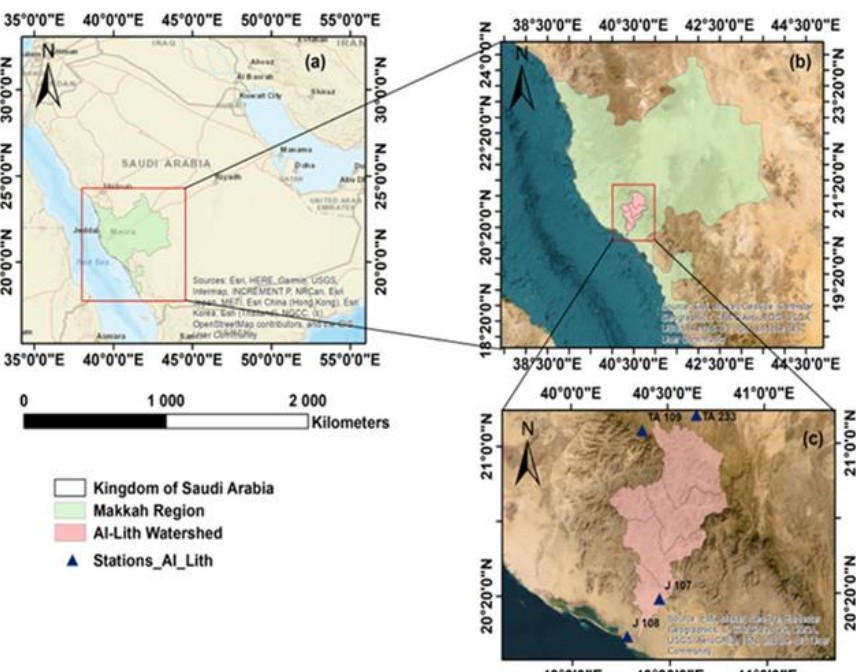

**Figure 1.** Geographical location of the Al-Lith Watershed: (**a**) Kingdom of Saudi Arabia; (**b**) Makkah Region; and (**c**) the Al-Lith Watershed.

The amount of precipitation in the Al-Lith Watershed varies with location and year. The average annual precipitation is 104.3 mm, with 145.2 mm, 167.7 mm, 56.7 mm, and 47.6 mm at stations TA-109, TA-233, J107, and J108, respectively. TA-109 and TA-233 are on the upstream side of the Al-Lith Watershed, whereas J107 and J108 are on the downstream

side, as shown in Figure 1c. The rainy season from November to January contributed to 55% of annual precipitation, while the dry season from June to August contributed only 15%.

## 3. Data and Methods

The methodology used in this study is categorized into; (i) meteorological data acquisition, (ii) satellite data retrieval, (iii) meteorological drought monitoring using SPEI, (iv) drought monitoring using remotely sensed drought indices (i.e., VCI, TCI, and VHI), (v) temporal and geospatial analyses of the above indices using GEE, and (vi) correlation between SPEI and remote sensing drought indices. The workflow for this study is shown in Figure 2.

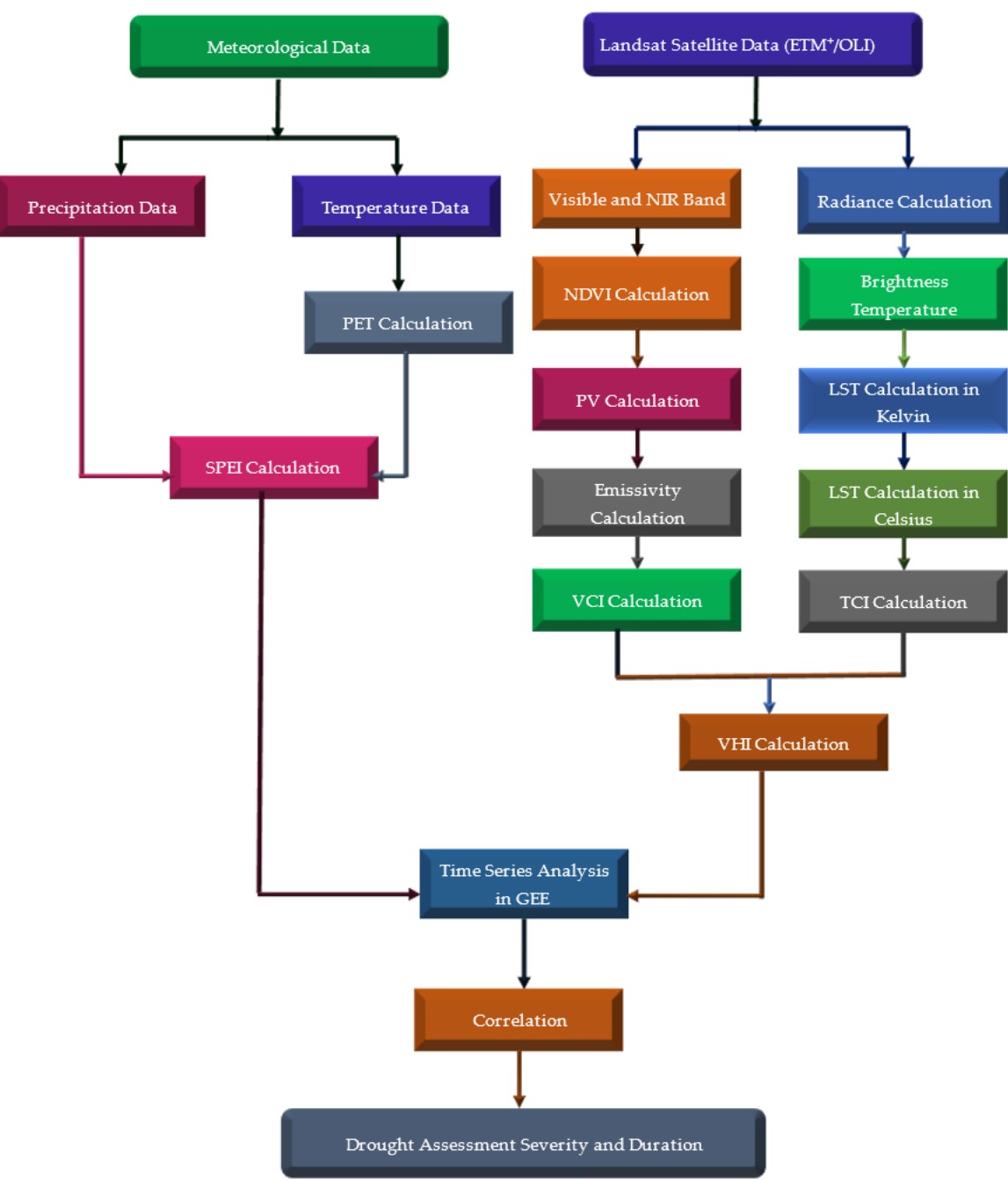

**Figure 2.** Methodological framework used in this study.

### 3.1. Meteorological Data Acquisition

This study utilized the daily precipitation and temperature data from the Ministry of Water Environment and Agriculture (MEWA). The data for four meteorological stations (J107, J108, TA109, and TA233, as shown in Figure 1) located in the Al-Lith watershed was collected spanning the period from 2001 to 2020.

### 3.2. Satellit Data Acquisitions

Landsat-7 and 8 Level 1 images at different dates were downloaded from the USGS Earth Explorer data portal between 2001 and 2020 for the study area. During the study period, Landsat-7 ETM+ data was acquired for the period of 2001–2012, while Landsat-8 OLI data was used for the period of 2013–2020. Images downloaded from the USGS Earth Explorer website were already georeferenced and projected in WGS UTM Zone 37 N for the Al-Lith watershed. However, no images were available for path 169 and row 046 on Earth Explorer, so the time series analysis for that specific location was done using Google Earth Engine. Table 1 shows the specifications, including sensors information, spatial and temporal resolutions, path, and rows for Landsat 7 and Landsat 8 satellites in Al-Lith watershed.

**Table 1.** Specification of Landsat Satellites over Al-Lith watershed.

| Satellite | Sensor | Spatial Resolution | Temporal Resolution | Paths | Row | Years |
|-----------|--------|--------------------|--------------------|-------|-----|-------|
| Landsat 7 | ETM+ | 30 m | 16days | 169 | 45 | 2001–2012 |
| Landsat 8 | OLI+ | 30 m | 16days | 169 | 46 | 2013–2020 |

### 3.3. Standard Precipitation Evapotranspiration Index (SPEI)

SPEI is the new and comprehensive drought index proposed by Vicentro et al. [6] and is used in this study to monitor meteorological drought. The SPEI is based on precipitation (P) and potential evapotranspiration (PET) and represents the fundamental calculation of the climate's water balance at various time scales. The PET can be calculated using many methods like Thornthwaite, Penman–Monteith, and Hargreaves. However, we used the Hargreaves equation in this study to calculate the PET because of less data requirement [49] suggested by many authors when solar radiation, relative humidity, and wind speed datasets are not available [50,51].

$$PET_i = 0.0135K_T(T + 17.78)(T_{max} - T_{min})^{0.5}R_a \qquad (1)$$

where $T$, $T_{max}$, and $T_{min}$ are the average, maximum, and minimum temperature in °C, respectively; $R_a$ is the extraterrestrial radiation (mm/day); and $K_T$ is an empirical coefficient ($K_T = 0.162$ for "interior" regions and $K_T = 0.19$ for coastal regions). Allen [52] also specified a criteria for calculating the observational coefficient as $K_T = 0.17$.

The monthly water balance (WB) equation is given by Equation (2), which is obtained by subtracting the calculated PET form monthly precipitation data:

$$WB_i = P_i - PET_i \qquad (2)$$

where $P$ is precipitation, and $i$ shows the particular month.

Mostly, the SPEI is computed by first standardizing the differences in precipitation and PET values using the log-logistic probability distribution function. The log-logistic distribution function is given below by Equation (3):

$$f(x) = \frac{\beta}{\alpha}\left(\frac{x - \gamma}{\alpha}\right)^{\beta - 1}\left[1 + \left(\frac{x - \gamma}{\alpha}\right)^{\beta}\right]^{-2} \qquad (3)$$

where $\alpha$ is the scale parameter, $\beta$ is the shape parameter, $\gamma$ is the beginning parameter, and x is the mean of the series of CWB values in each period. Then SPEI can be calculated from [6]

$$\text{SPEI} = W - \frac{C_o + C_1 W + C_2 W^2}{1 + d_1 W + d_2 W^2 + d_3 W^3} \tag{4}$$

where $W = \sqrt{-2Ln(P)}$ for $p \leq 0.5$, and P is the probability of exceeding a determined WB value. The constants are $C_0 = 2.515517$, $C_1 = 0.802853$, $C_2 = 0.010328$, $d_1 = 1.432788$, $d_2 = 0.189269$, and $d_3 = 0.001308$ [6].

SPEI can be estimated at different time scales (1, 3, 6, and 12 months) at each station. SPEI-1 is calculated by taking the monthly precipitation and temperature. SPEI-3 is calculated by taking the mean of the three months (moving averaging of three-month precipitation and temperature inputs). Similarly, the remaining 6- and 12-month indices can be calculated. SPEI-1 is useful to study the short-term variations in drought frequency and severity, SPEI-3 and SPEI-6 are usually used to monitor the seasonal variations in drought, while SPEI-12 is useful to study the annual trend of drought. How severe a drought is can be estimated using the index, which compares actual precipitation to the amount of water lost through evaporation and transpiration, over a given period of time.

The severity of drought determined by the numeric values of SPEI, which is divided into different categories following McKee et al. [5] and Vicentro et al. [6] and are shown in Table 2.

**Table 2.** Division of drought severity based on SPEI values (after [6]).

| SPEI | Categories |
|---|---|
| >2 | Extremely wet |
| 1.50 to 1.99 | Severely wet |
| 1.00 to1.49 | Moderately wet |
| −0.99 to 0.99 | Nearly Normal |
| −1.49 to −1.0 | Moderately drought |
| −1.99 to −1.5 | Severe drought |
| <−2 | Extreme drought |

*3.4. Remote Sensing-Derived Indices*

3.4.1. Vegetation Condition Index (VCI)

The Vegetation Condition Index (VCI) is considered a step forward in analyzing vegetation conditions, particularly in non-homogeneous environments [53]. VCI can extract the impact of weather on plants while removing the ecosystem signal from NDVI [54] and is defined as follows:

$$VCI_{ij}^k = 100 * \frac{NDVI_{ij}^k - \min\left(NDVI_i^k\right)}{\max\left(NDVI_i^k\right) - \min\left(NDVI_i^k\right)} \tag{5}$$

where $VCI_{ij}^k$ and $NDVI_{ij}^k$ indicate VCI and NDVI values at pixel *k*, in *i*-th month, for the year *j*. The $\max\left(NDVI_i^k\right)$ and $\min\left(NDVI_i^k\right)$ shows the maximum and minimum values of NDVI in the period. The VCI values range from 0 to 100, where VCI values below 40 indicate drought conditions in the area (presented in Table 3) [45,55–57]. NDVI is calculated using the red and near-infrared (NIR) bands and is given by [19].

$$\text{NDVI} = \frac{\text{NIR} - \text{RED}}{\text{NIR} + \text{RED}} \tag{6}$$

**Table 3.** Different drought categories for RSDIs (modified from [58]).

| VHI/VCI/TCI Values | Drought Class |
|---|---|
| 0 to 10 | Extreme Drought |
| 10 to 20 | Severe Drought |
| 20 to 30 | Moderate Drought |
| 30 to 40 | Mild Drought |
| More than 40 | No Drought |

### 3.4.2. Temperature Condition Index (TCI)

The Temperature Condition Index (TCI) considers that a drought occurrence will reduce soil moisture and increase land surface thermal stress, i.e., there will be a higher land surface temperature (LST) in drought periods compared to normal ones. A high LST during the growing season of crops implies unfavorable or drought conditions, whereas a low land surface temperature suggests predominantly favorable conditions [31]. TCI is connected to the responsiveness of vegetation to any unfavorable changes in temperature. The following expression demonstrates the calculation of TCI [20]:

$$TCI_{ij}^k = 100 \; * \; \frac{\max\left(LST_i^k\right) - LST_{ij}^k}{\max\left(LST_i^k\right) - \min\left(LST_i^k\right)} \tag{7}$$

where $TCI_{ij}^k$ and $LST_{ij}^k$ indicate TCI and LST values at $k$ pixel, in $i$-th month, for the year $j$. The $\min\left(LST_i^k\right)$ and $\max\left(LST_i^k\right)$ shows the minimum and maximum values of LST in the period. $TCI_{ij}^k$ values vary from 0 to 100, indicating stress to good thermal conditions of the vegetation, as shown in Table 3. To calculate LST the following steps are used [59].

$$L_\lambda = M_L \cdot Q_{CAL} + A_L \tag{8}$$

where $L_\lambda$ is the Top of Atmosphere (TOA) spectral radiance $\left(Wm^{-2}sr^{-1}\,mm^{-1}\right)$, $M_L$ is the band-specific multiplicative rescaling factor from the metadata, $A_L$ is the band-specific additive rescaling factor from the metadata, and $Q_{CAL}$ is the quantized and calibrated standard product pixel values (DN). All of these variables can be retrieved from the metadata file of Landsat 8 data.

TOA to brightness temperature can be calculated using the following Equation (9) [60,61]

$$BT = \left(\frac{K_2}{\left(\ln\left(\frac{K_1}{L}\right) + 1\right)}\right) - 273.15 \tag{9}$$

where K1 and K2 are band specific thermal conversion constants and can be found from metadata of the image.

The proportion of vegetation (Pv) is calculated by using the following Equation (10) [62]

$$Pv = \text{square}\left(\frac{NDVI - NDVI_{\min}}{NDVI_{\max} - NDVI_{\min}}\right) \tag{10}$$

The emissivity is calculated using the following Equation (11) [63,64]

$$\varepsilon = 0.004 \; * \; P_v + 0.986 \tag{11}$$

Finally, the LST was derived using the following Equation (12) [65]

$$LST = \frac{BT}{\left(1 + \left[\left(\lambda BT / \rho\right) \ln \varepsilon\right]\right)} \tag{12}$$

where $\lambda$ is the effective wavelength (10.9 mm for band 10 in Landsat 8 data) and $\varepsilon$ is the emissivity.

$$\rho = h\frac{c}{\sigma} = 1.438 \times 10^{-2} \text{mK} \tag{13}$$

where $\sigma$ is the Boltzmann constant ($1.38 \times 10^{-23}$ J/K), $h$ is Plank's constant ($6.626 \times 10^{-34}$ Js), and $c$ is the velocity of light in a vaccum ($2.998 \times 10^8$ m/s).

### 3.4.3. Vegetation Health Index (VHI)

The following equation is used to compute the VHI index, which is the weighted sum of VCI and TCI and is a useful source of information regarding the stress on vegetation by droughts. Gidey et al. [55] reported that the coefficient of the VHI equation was kept at 0.5 due to a lack of more precise information on the influence of VCI and TCI on the VHI.

$$\text{VHI} = a\text{VCI} + (1 - a)\text{TCI} \tag{14}$$

where $a = 0.5$ (the same contribution of VCI and TCI). Droughts based on VHI are classified into five categories, following to the recommendations of Kogan [53]. Table 3 shows different drought conditions based on VCI, TCI, and VHI values.

### 3.5. Pearson Corerlation Coefficient

The linear relationships between RSDIs and the meteorological drought index (SPEI) were evaluated on a 1-, 3-, 6-, and 12-month time scales in Wadi Al-Lith from 2001 to 2020 by utilizing the Pearson Correlation Coefficient (CC). The Standardized Anomaly Index (SAI) was used to discover anomalies in RSDIs by calculating a standardized deviation from the long-term mean. Then, the anomalies of RSDIs are compared with SPEI at different time scales to analyze the correlation between them. The following equation is used to calculate the SAI [66,67].

$$\text{SAI}_i = \frac{x_i - \overline{x}}{\sigma} \tag{15}$$

where $x_i, \overline{x}$, and $\sigma$ represents the values of RSDIs at any month, the long-term mean, and the standard deviation, respectively.

## 4. Results

### 4.1. Evaluation of Drought Indices

The meteorological drought index (SPEI) at different time scale, i.e., 1-, 3-, 6-, and 12-months and RSDIs (i.e., VCI, TCI, and VHI) are used to analyze the drought severity/condition on both spatial and temporal scale from 2001 to 2020 in the Al-Lith watershed.

### 4.1.1. SPEI

Figure 3 shows the SPEI time series at different time scales (i.e., 1-, 3-, 6-, and 12-months) at stations J107, J108, TA-109, and TA-233 from 2001 to 2020 in the Al-Lith watershed. Figure 3a shows that more severe and extreme drought events are observed by SPEI-3, followed by SPEI-12 and SPEI-6, whereas SPEI-1 shows moderate to severe drought conditions. However, high fluctuations are observed in SPEI-1 and SPEI-3 time series compared to SPEI-6 and SPEI-12. Moreover, the time series analyses depict severe to extreme drought events in 2002, 2003, 2007, 2011, 2015, 2017, and 2019 (Figure 3a). Overall, the time series plot shown in Figure 3b demonstrates more severe to extreme drought events in 2003, 2007, 2009, 2012, 2016, 2019, and 2020. Further, drought at stations TA109 (Figure 3c) and TA233 (Figure 3d) are more severe at 3- and 12-month time scales followed by SPEI-6 and SPEI-1. Extreme and severe drought events at station TA109 are observed in 2002, 2007, and 2012. Likewise, severe to extreme drought events at station TA233 are observed in 2005, 2006, 2009, 2012, and 2019. Overall, the results (Figure 3a–d) demonstrate significant numbers of extreme and severe drought events in 2002, 2007, 2009, 2012, 2015, and 2019 in the Al-Lith watershed.

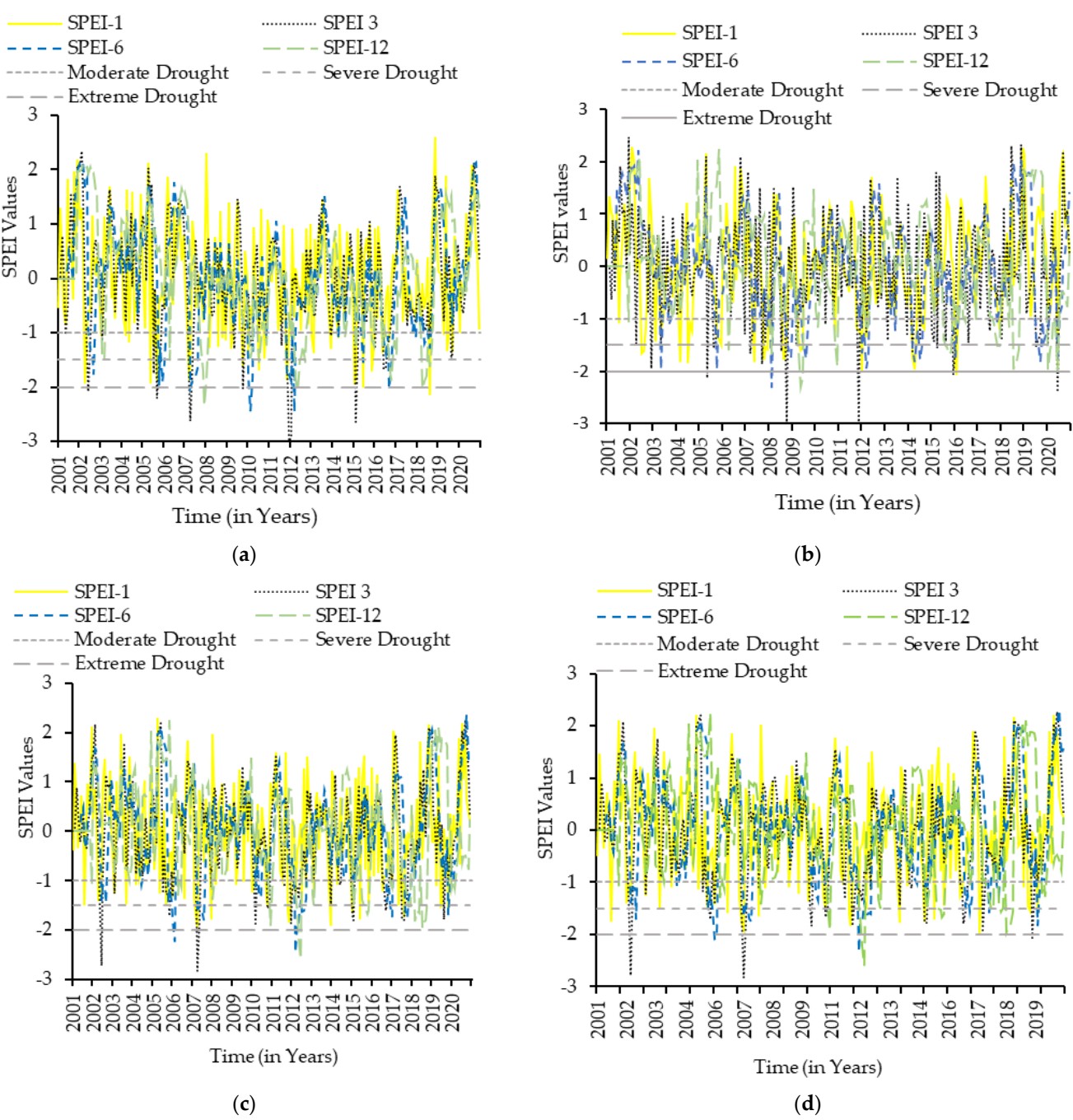

**Figure 3.** Time series analysis at four stations of the AlLith watershed on different time scales of SPEI (1-, 3-, 6-, and 12-month) at (**a**) J107, (**b**) J108, (**c**) TA109, and (**d**) TA233.

4.1.2. VCI

Figures 4 and 5 show the spatial and temporal distribution of VCI from 2001 to 2020 in the Al-Lith Watershed, which is calculated from the NDVI, shown in Figure A1 in Appendix A. The spatial distribution maps of VCI (Figure 4) indicate that 2001, 2002, 2004, 2007, 2008, 2010, and 2011 are the extreme drought years in the study period. In contrast, more wet conditions (relatively less drought) are observed in 2005, 2006, 2009, and 2016. Furthermore, it should be noted that drought is relatively more severe in northern areas of the Al-Lith watershed compared to southern areas.

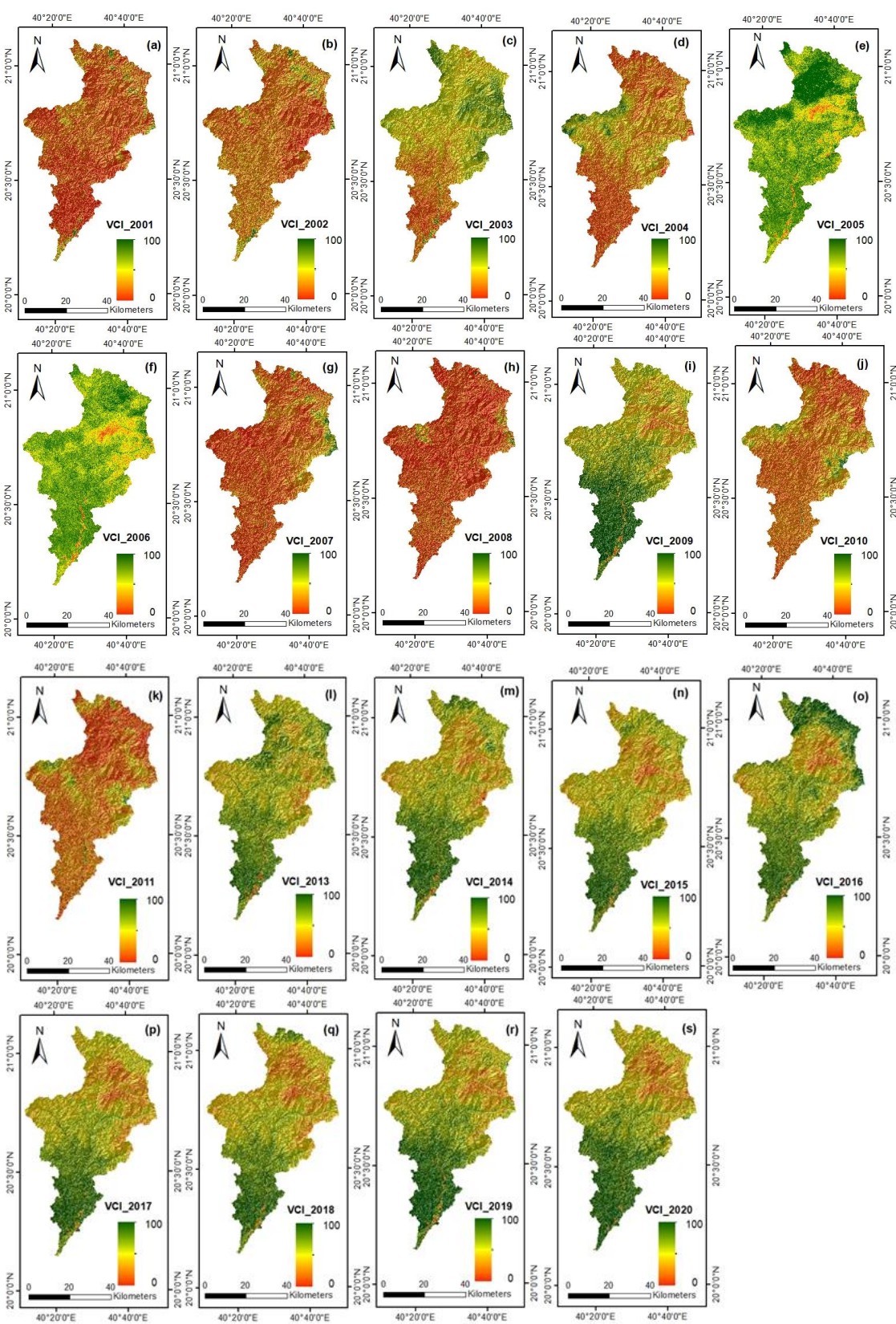

**Figure 4.** (**a–s**) Spatial distribution of VCI in Al-Lith Watershed retrieved from Landsat Satellites for the period of 2001–2020.

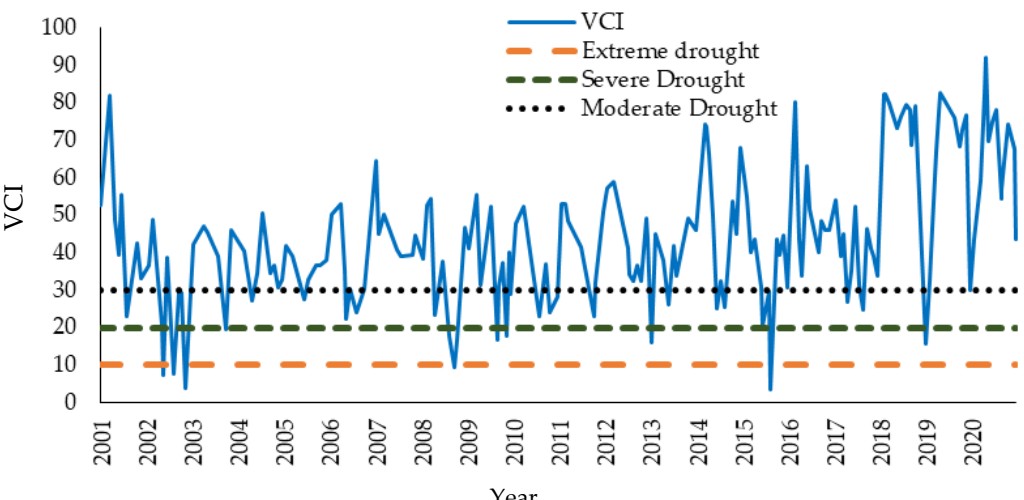

**Figure 5.** Time series plot of VCI in Al-Lith Watershed extracted using Google Earth Engine and Landsat Satellite datasets.

The time series of VCI extracted using the GEE (shown in Figure 5) illustrate that the mean VCI values range from 20.3 to 64.7 during the study period. According to the VCI time series plot, the Al-Lith watershed experienced minimum VCI values (severe and extreme drought events) in 2001, 2002, 2003, 2007, 2008, and 2010, in the first decade. Similarly, the minimum VCI values in the second decade are observed in 2013, 2015, 2016, and 2019. In other words, the VCI time series demonstrates that 2001, 2002, 2008, 2015, and 2018 are the extreme drought years.

### 4.1.3. TCI

Figures 6 and 7 show the spatial distribution and time series plot of TCI in the Al-Lith watershed spanning a period of 2001–2020 retrieved from Landsat satellites and GEE, respectively. As shown in Figure 6, the minimum TCI values are observed in 2001, 2003, 2005, 2006, 2009, 2011, 2013, 2015, 2017, 2018, and 2020. The temperature in the study area is increasing over time (Figure A2 in the Appendix A), therefore, the results show more severe TCI in the last decade compared with first decade. Since the Al-Lith watershed is located in a hyper-arid region, where the maximum temperature reaches above 40 °C and annual precipitation is usually less than 10 mm, the TCI will have a significant impact and contribution to VHI index. TCI has a similar spatial distribution trend to that of VCI, i.e., TCI is more severe in northern areas of Al-Lith compared with southern areas, except for a few years (2002, 2004, 2010, and 2014, as shown in Figure 6).

The spatial distributions of VCI and TCI are different in some periods, such as the year 2001 (Figures 4a and 6a), which is mainly caused by different views of these indices in describing the drought. VCI is calculated from NDIV and considers only the vegetation factor, while TCI is calculated from LST and is an integrated results of many factors, including vegetation, precipitation, topography, elevation, soil, and meteorology.

The time series plot of TCI for the Al-Lith watershed is shown in Figure 7. TCI values show that the Al-Lith watershed experienced frequent severe and extreme drought events in 2001, 2002, 2004, 2005, 2007, 2008, 2013, 2014, 2015, 2017, 2019, and 2020. Overall, the mean TCI values across different years ranges from 13.79 to 89.6 in the study area.

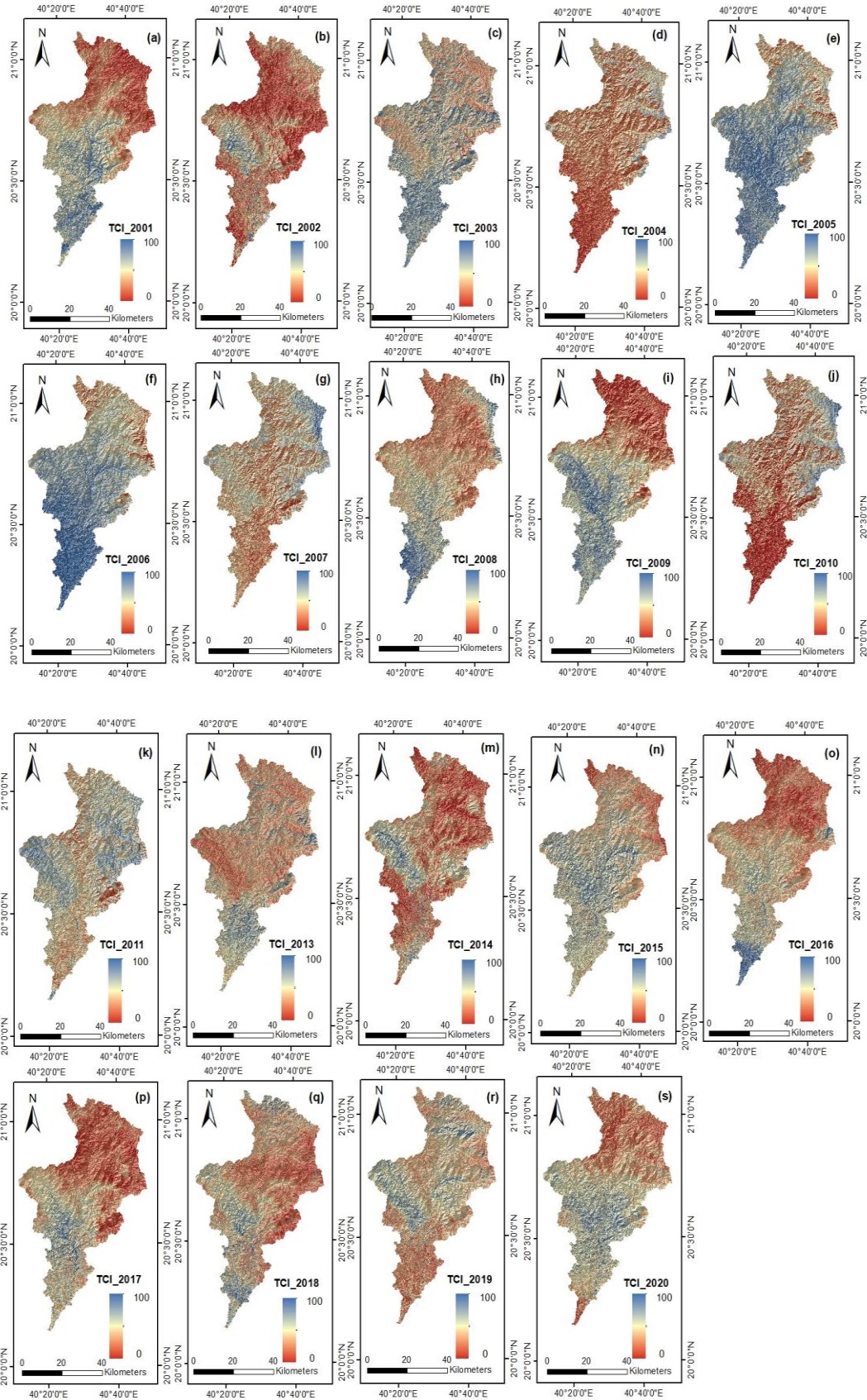

**Figure 6.** (**a**–**s**) Spatial distribution of TCI in Al-Lith Watershed retrieved from Landsat Satellites for the period of 2001–2020.

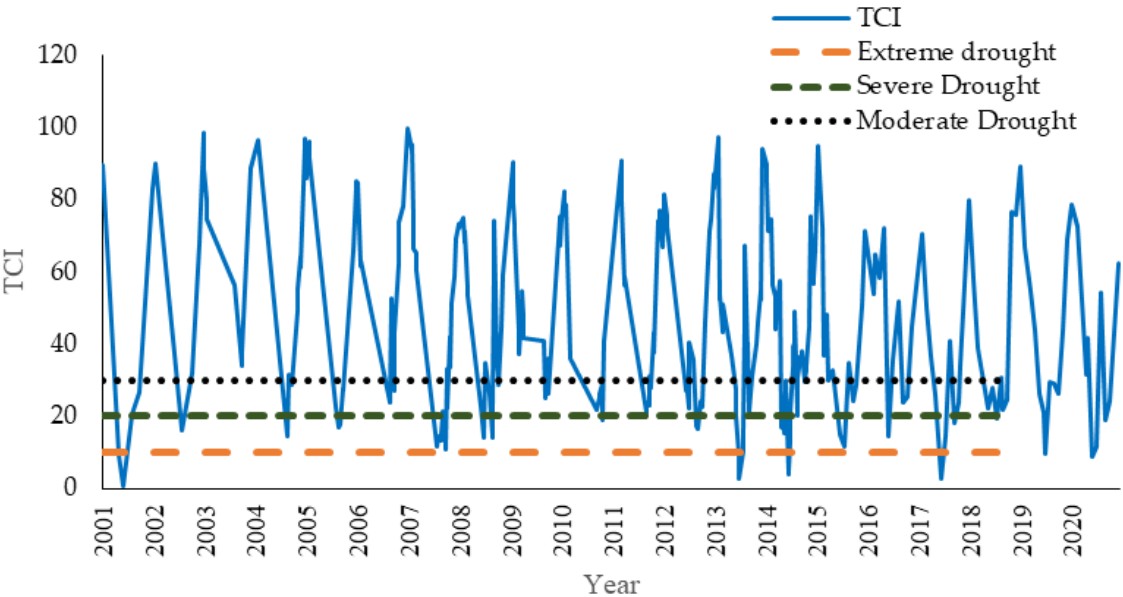

**Figure 7.** Time series plot of TCI in Al-Lith Watershed extracted using Google Earth Engine and Landsat Satellite datasets.

### 4.1.4. VHI

The spatial distribution and time series plot of VHI across the Al-Lith watershed extracted from Landsat satellite imageries and GEE, respectively, is shown in Figures 8 and 9. Figure 8 depicts that 2001, 2002, 2004, 2007, 2008, 2010, 2015, 2018, and 2019 are the severe drought years across the Al-Lith watershed. However, it should be noted that drought estimates using VHI depict that drought is more severe in the first decade than the second one. Furthermore, the figure also shows that drought is more severe in the downstream region rather than upstream region of the Al-Lith watershed. Since VHI is a more comprehensive drought index than VCI and TCI, the VHI shows that downstream areas of the Al-Lith watershed is more vulnerable to drought, and thus, it is advised to devise robust mitigation plans to encounter the adverse impacts of drought on available water reserves and agriculture in the downstream region of the Al-Lith watershed.

The time series plot of the VHI, extracted from GEE (shown in Figure 9), illustrates that the mean VHI values range from 21.03 to 61.05 during the entire study period. According to the VHI time series plot, the Al-Lith watershed experienced the minimum VHI values (severe and extreme drought events) in 2002, 2004, 2007, 2008, 2010, 2012, 2013, 2014, 2015, and 2017–2020. It is worth mentioning that intensity and frequency of severe drought events are significantly increased in the second decade, particularly after 2017.

The maximum, minimum, and average values of VCI, TCI, and VHI across the Al-Lith watershed on an annual scale from 2001 to 2020 are shown in Table 4. Table 4 shows extreme droughts in 2002, 2008, 2015 while moderate droughts in 2001, 2003, 2006, 2009, 2010, 2013, and 2018 for VCI. Similarly, the extreme (severe) drought events for TCI are observed in 2001, 2013, 2014, 2017, 2019, and 2020 (2002, 2004, 2005, 2007, 2008, 2010, 2012, 2015, 2016, and 2018). On the other hand, no extreme drought events are observed for VHI whereas severe drought is observed in 2004, 2007, 2012, 2013, and 2017–2020).

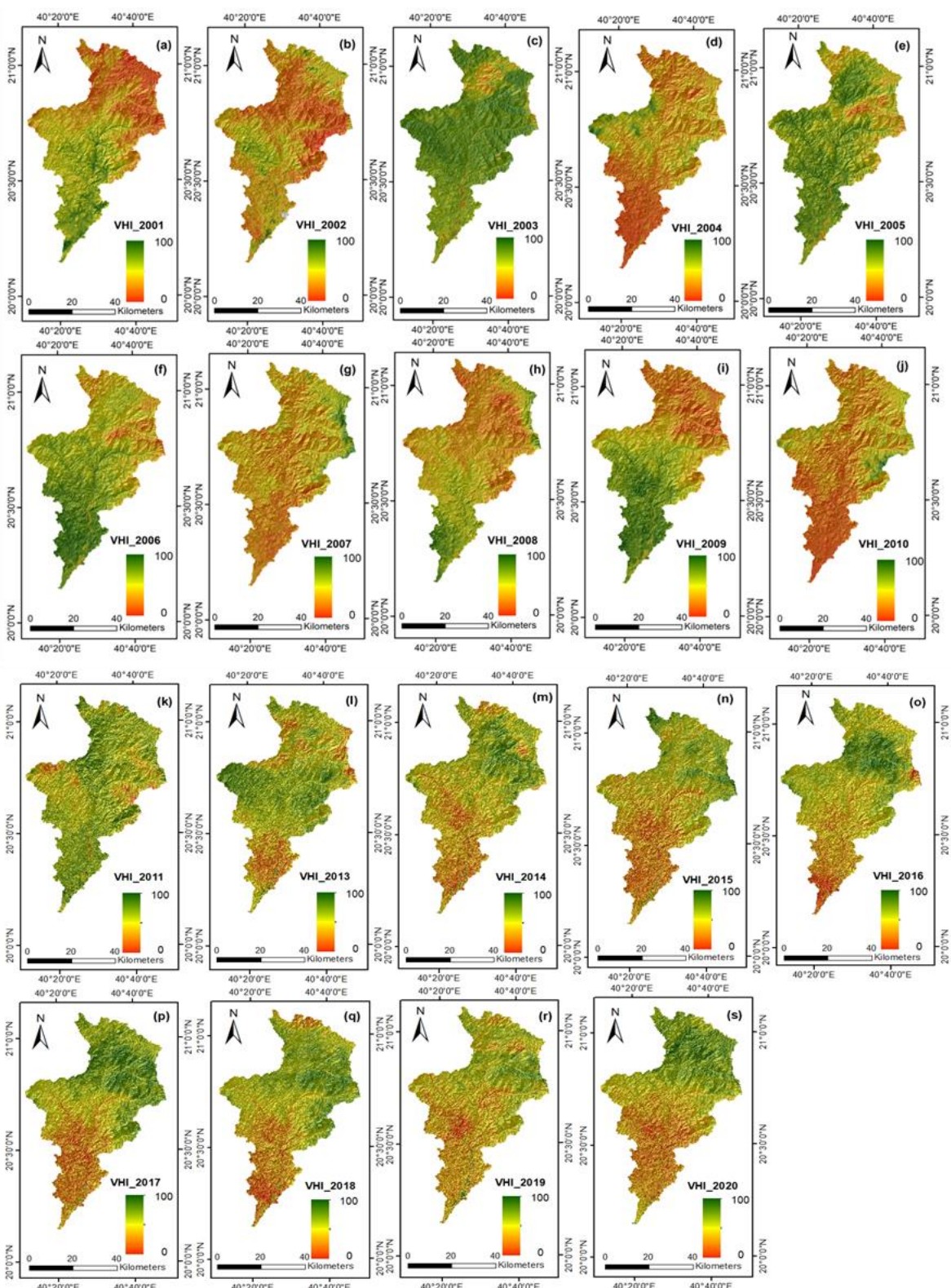

**Figure 8.** (**a**–**s**) Spatial distribution of VHI in Al-Lith Watershed retrieved from Landsat Satellites for the period of 2001–2020.

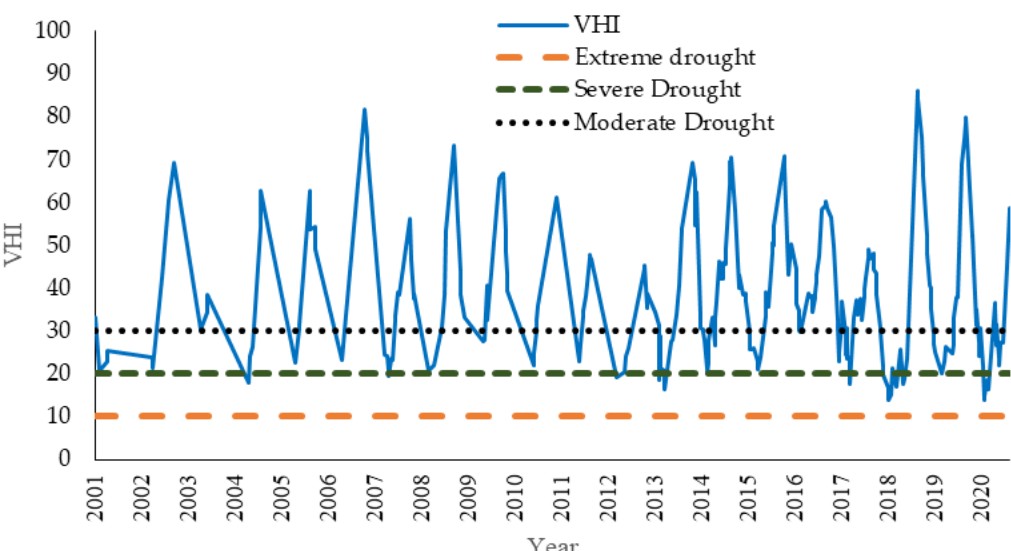

**Figure 9.** Time series plot of VHI in Al-Lith watershed extracted using Google Earth Engine and Landsat Satellite datasets.

**Table 4.** Minimum, Maximum, and Mean values of VCI, TCI, and VHI (2001–2020).

| Year | VCI | | | TCI | | | VHI | | |
|---|---|---|---|---|---|---|---|---|---|
| | Min | Max | Average | Min | Max | Average | Min | Max | Average |
| 2001 | 19.68 | 93.97 | 56.83 | 0.52 | 89.56 | 45.04 | 20.82 | 33.25 | 27.03 |
| 2002 | 3.97 | 61.56 | 32.76 | 11.72 | 89.96 | 50.84 | 21.40 | 60.44 | 40.92 |
| 2003 | 19.66 | 53.78 | 36.72 | 33.92 | 98.52 | 66.22 | 30.33 | 69.27 | 49.80 |
| 2004 | 25.67 | 50.46 | 38.06 | 14.52 | 96.92 | 55.72 | 18.05 | 62.71 | 40.38 |
| 2005 | 25.43 | 46.15 | 35.79 | 16.69 | 95.98 | 56.33 | 22.65 | 62.72 | 42.68 |
| 2006 | 17.82 | 58.64 | 38.23 | 23.71 | 99.66 | 61.68 | 23.18 | 54.33 | 38.76 |
| 2007 | 26.00 | 64.52 | 45.26 | 10.79 | 95.35 | 53.07 | 19.54 | 81.81 | 50.68 |
| 2008 | 9.43 | 60.51 | 34.97 | 13.86 | 74.83 | 44.34 | 20.73 | 56.22 | 38.48 |
| 2009 | 16.68 | 55.61 | 36.15 | 25.00 | 90.24 | 57.62 | 27.42 | 73.19 | 50.30 |
| 2010 | 19.22 | 57.58 | 38.40 | 19.05 | 82.41 | 50.73 | 21.84 | 66.87 | 44.35 |
| 2011 | 23.02 | 52.94 | 37.98 | 20.87 | 90.94 | 55.91 | 22.77 | 61.23 | 42.00 |
| 2012 | 26.26 | 60.52 | 43.39 | 16.34 | 87.25 | 51.80 | 19.11 | 25.87 | 22.49 |
| 2013 | 16.03 | 49.03 | 32.53 | 2.58 | 97.19 | 49.88 | 16.45 | 53.90 | 35.17 |
| 2014 | 25.19 | 74.14 | 49.67 | 4.00 | 89.60 | 46.80 | 20.09 | 70.41 | 45.25 |
| 2015 | 3.48 | 54.69 | 29.09 | 11.43 | 94.88 | 53.15 | 21.03 | 58.64 | 39.83 |
| 2016 | 33.81 | 80.14 | 56.98 | 13.70 | 88.10 | 50.90 | 29.60 | 70.72 | 50.16 |
| 2017 | 22.56 | 54.46 | 38.51 | 2.49 | 71.84 | 37.17 | 17.72 | 60.16 | 38.94 |
| 2018 | 10.74 | 85.96 | 48.35 | 16.37 | 84.06 | 50.21 | 14.01 | 86.04 | 50.03 |
| 2019 | 28.06 | 84.65 | 56.35 | 9.40 | 96.63 | 53.02 | 19.99 | 75.30 | 47.65 |
| 2020 | 33.25 | 94.75 | 64.00 | 8.87 | 78.69 | 43.78 | 13.98 | 79.95 | 46.96 |
| 2001–2020 | 20.30 | 64.70 | 42.50 | 13.79 | 89.63 | 51.71 | 21.04 | 63.15 | 42.09 |

### 4.2. Correlation between SPEI, VCI, TCI, and VHI

The Pearson Correlation Coefficient (CC) between the meteorological drought index (SPEI) and RSDIs (VCI, TCI, and VHI) is calculated in the Al-Lith watershed from 2001 to 2020 and presented in Table 5. TCI and VHI have shown good agreement with SPEI in the study period, where a CC value of 0.64 is observed between the two indices, as illustrated in Table 5. VHI and VCI show reasonable agreement with each other where a CC value of 0.51 is observed between the two indices. However, the correlation drops significantly to a CC value of 0.39 between VCI and TCI.

**Table 5.** Correlation matrix between Remote sensing and meteorological drought indices based on GEE) and stations data over the Al-Lith watershed from 2001–2020.

|  | **VHI** | **TCI** | **VCI** | **SPEI-1** | **SPEI-3** | **SPEI-6** | **SPEI-12** |
|---|---|---|---|---|---|---|---|
| VHI | 1.00 | 0.64 | 0.51 | 0.47 | 0.57 | 0.67 | 0.72 |
| TCI | 0.64 | 1.00 | 0.39 | 0.37 | 0.42 | 0.61 | 0.60 |
| VCI | 0.51 | 0.39 | 1.00 | 0.24 | 0.33 | 0.45 | 0.53 |

Further, the analyses are further extended and the correlation between SPEI and RSDIs is investigated at different time scales (1-, 3-, 6-, and 12-month). The results (Table 5) show that SPEI-12 strongly correlates with VHI, where a CC value of 0.72 is observed between VHI and SPEI-12. VHI also has a strong correlation with SPEI-6 (CC equals 0.67), which drops significantly to 0.47 with SPEI-1. In contrast to VHI, a lower correlation is observed between VCI and SPEI. The maximum and minimum CC is observed between VCI/SPEI-12 and VCI/SPEI-1, respectively. TCI shows a moderate correlation with SPEI, where the maximum correlation (with CC of 0.61) is observed between TCI/SPEI-6, followed by TCI/SPEI-6.

From Table 5, the correlation between VHI/VCI/TCI and SPEI increases with time scales of SPEI. In the calculations of VHI, TCI, and VCI, monthly data are used. However, the vegetation growth is related with wet/drought conditions of both the present month and previous months of vegetation growth, or even months before vegetation growth. Consequently, vegetation growth regime and vegetation index are more closely related with wet/drought conditions of longer time scales, and VHI, TCI, and VCI, closely related with vegetation index, all tend to have a stronger correlation with SPEI over a longer time frame.

## 5. Discussion

Drought catastrophes not only significantly affect the current agricultural productivity, human life, and economic development, but can potentially worsen climate conditions and exacerbate land desertification for a long period. Therefore, it is of utmost importance to investigate the causes of drought disasters, conduct in-depth analyses of their spatiotemporal characteristics, and employ scientific strategies for their prevention and management. The research on the disaster caused by drought is still in its infancy since researchers typically only look at one drought index at a time rather than the cumulative effects of numerous drought indices. Unlike previous studies, this study examined the spatial and temporal changes of various drought indices while taking precipitation, vegetation index, and surface temperature into account in the Arid region of KSA from 2001–2020.

The current study is an attempt to evaluate and correlate in situ meteorological drought (SPEI) and remote sensing-retrieved drought indices (VCI, TCI, and VHI) in the western region of KSA (the Al-Lith watershed) from 2001 to 2020. RSDIs are retrieved from Landsat 7 and Landsat 8 satellite datasets, and the GEE platform. The evaluation of in situ SPEI against the RSDIs is very time consuming in KSA because of the very limited availability of in situ climate data and the sparse distribution of weather stations. The climate data from weather stations is usually incomplete, with several missing observations and significant uncertainties [10,68,69]. The variations in temperature, precipitation, evaporation, etc., are hardly captured by the in situ data owing to the limited stations, its sparse distribution, and incomplete data [70]. Temperature in KSA are rising, droughts are getting worse, and nearly 70% of the country is affected by severe and frequent drought events [69,71]. For instance, the Al-Lith watershed has experienced severe drought events over the past 20 years, consistent with numerous regional and global assessments [25,44,69,72–75]. Therefore, it is of utmost importance to precisely estimate the increasing severity of drought using various indices, especially the RSDIs, to address the data scarcity issues and its contribution to the uncertainties in drought estimation.

GEE and the long-term continuous observations from the Landsat satellite datasets enabled us to compare the RSDIs (VCI, TCI, and VHI) with meteorological drought (SPEI). Many studies used GEE and Landsat satellite data to monitor droughts in different regions worldwide [63,72–74] by using VCI, TCI, and VHI. VHI is used to monitor agricultural drought and is suggested by many authors for drought monitoring [20,31,44,58,75]. According to Choi et al. [76], VHI reflects the anomalies in both vegetation cover and temperature. Table 5 illustrates that VHI showed good agreement with VCI, but the correlation is slightly weaker with TCI. VCI and TCI also play a critical role in drought monitoring because both indices depend on ecological conditions and weather-induced changes, respectively. VCI depends on the maximum and minimum NDVI, which reflects the regional vegetation health. On the other hand, TCI is based on the minimum and maximum of LST, i.e., more dependent on temperature. The results of VCI, TCI, and VHI indicate that the watershed experienced severe drought in 2001, 2003, 2007, 2008, 2013, 2014, 2017, and 2019, among other studied years.

Table 5 shows that SPEI and VHI show a strong correlation with each other in the study area, while the correlation is moderate to weak between SPEI/TCI and SPEI/VCI, respectively. Gidey et al. [44] found a very strong correlation between the meteorological drought index (SPI) and RSDIs (VCI/TCI/VHI) while studying the statistical relationship between meteorological and RSDIs in Northern Ethiopia. Del-Toro-Guerrero et al. [66] reported a very strong correlation between SPI and VHI annually while studying the surface reflectance derived indices for drought monitoring. Almeida-Ñauñay et al. [77] studied the impact of the meteorological and RSDIs in semi-arid Mediterranean Grass Land using station and satellite remote sensing data. They reported a good correlation between VHI and SPEI, which support the findings of this study.

Overall, this study utilized Landsat satellite data and generated time series analysis on GEE to investigate the spatial and temporal distribution of drought in the hyper-arid and data scarce regions of KSA. This study combined the meteorological and remote sensing indices for monitoring drought in an arid region. The results showed that VHI is a more robust drought index, showing a good correlation with station-based meteorological drought index (SPEI), and thus is more robust to represent drought over KSA. In other words, RSDIs have the capability to represent the drought condition comparatively well compared with station-based estimated drought. This research gives a clear picture of the drought assessment, which can be helpful for the policymakers, researchers, and government to take proper action for drought mitigation.

## 6. Conclusions

This study assessed the spatial and temporal distribution of droughts, including meteorological drought estimated through SPEI (SPEI-1, SPEI-3, SPEI-6, and SPEI-12) and remote sensing-retrieved drought indices (RSDIs, including VCI, TCI, and VHI). Moreover, the SPEI and RSDIs are correlated with each other at different time scales (i.e., 1-, 3-, 6-, and 12-month) using the Pearson correlation coefficient. The Standardized Anomaly Index (SAI) was used to calculate the anomalies for RSDIs and compared with SPEI at 1-, 3-, 6-, and 12-month time scales. The analyses are carried out over the hyper-arid region of KSA having limited in situ weather stations from 2001 to 2020. Our major findings are stated below.

(1) SPEI results showed that significant number of severe and extreme drought events are observed in 2002, 2007, 2009, 2012, 2015, and 2019. High fluctuations in drought severity are observed at smaller time scales (i.e., SPEI-1 and SPEI-3) compared with SPEI-6 and SPEI-12. However, interesting observations are observed at station J108 where drought is more severe at smaller time scales (SPEI-1) than larger times scales. Overall, the results showed significant regional variations in drought severity owing to regional changes in climate.

(2) VCI showed that northern areas of the Al-Lith watershed are prone to drought, particularly in 2001, 2002, 2004, 2007, 2008, 2010, and 2011. The spatial distribution of

VCI depicted extreme drought events in 2001, 2002, 2004, 2007, 2008, 2010, and 2011. VCI ranged from 20.3 to 64.7, where extreme (severe) drought events were observed in 2002, 2008, and 2018 (2001, 2003, 2006, 2009, 2010, 2013, and 2018).

(3) Based on the results of TCI, the Al-Lith watershed was prone to droughts in 2001, 2003, 2005, 2006, 2009, 2011, 2013, 2015, 2017, 2018, and 2020. Droughts were more intense in the second decade compared to the first decade. Time series plot of TCI showed that TCI values range from 10.60 to 91.34, where minimum TCI values are frequently observed in the last decade. Overall, the Al-Lith watershed was vulnerable to droughts in 2001, 2002, 2004, 2005, 2007, 2008, 2013, 2014, 2015, 2017, 2019, and 2020.

(4) VHI results depicted severe drought events in 2001, 2002, 2004, 2007, 2008, 2010, 2015, 2018, and 2019. The values of VHI ranged from a minimum of 19.47 to a maximum of 64.22. The VHI time series depicted severe and extreme drought events in 2002, 2004, 2007, 2008, 2010, 2012, 2013, 2014, 2015, and 2017–2020.

(5) The correlation analyses showed that VHI has a good correlation with VCI, with an average CC value of 0.51. On the other hand, a minimum correlation with CC value of 0.39 is observed between VCI and TCI. Highest correlation among RSDIs is observed between VHI and TCI, with a CC value of 0.64.

(6) The correlation between SPEI (at 1-, 3-, 6-, and 12-month) and RSDIs showed a good agreement between VHI/SPEI-12 and VHI/SPEI-6 with average CC values of 0.74 and 0.67, respectively. The correlation got weaker at smaller time scales, i.e., the average CC value between VHI/SPEI-3 and VHI/SPEI-1 are 0.42 and 0.37. A lower correlation of SPEI is observed with VCI, where the maximum CC value of 0.52 is estimated between VCI/SPEI-12 followed by 0.36 between VCI/SPEI-6. Further, a moderate correlation is observed between TCI and SPEI.

To conclude, droughts were monitored both from in situ data acquired from MEWA and remote sensing techniques. The comparison and correlation between in situ drought index (SPEI) and RSDIs indicated that RSDIs are accurate enough to represent drought conditions in hyper-arid regions like KSA. Contrasting results are observed for VCI and TCI, i.e., more severe droughts in the south than north as well as more severe droughts were observed in the first decade than the second because of interpolation and time series data, which is averaged for the whole Al-Lith watershed which does not represent the real scenario. Moreover, the remote sensing technology is found to be very useful in regions with limited weather stations and data availability. Therefore, the findings of this study are critical to our understanding of the nature of droughts in arid and hyper-arid regions. Prospective researchers will find useful information in the study's findings for resolving local and regional drought issues and in devising plans for drought mitigation.

**Author Contributions:** Conceptualization, J.B. and K.U.R.; methodology, J.B., K.U.R. and S.S.; software, N.E. and K.M.A.; validation, N.E. and K.M.A.; formal analysis, N.E. and K.U.R.; investigation, N.E. and K.M.A.; resources, N.E. and J.B.; data curation, N.E. and K.U.R.; writing—original draft preparation, N.E. and K.M.A.; writing—review and editing, J.B., K.U.R. and S.S.; visualization, N.E. and K.U.R.; supervision, J.B. and K.U.R.; project administration, J.B. and S.S.; funding acquisition, S.S. and K.U.R. All authors have read and agreed to the published version of the manuscript.

**Funding:** This research was funded by the Natural National Science Foundation of China (Grant Numbers 51839006 and 52250410336), China Postdoctoral Science Foundation (Grant number 2022M721872), and the Shuimu Scholar Program of Tsinghua University (Grant Number 2020SM072).

**Data Availability Statement:** Data will be available on request to the first two authors.

**Acknowledgments:** The authors acknowledge Ministry of Water Environment and Agriculture (MEWA) for providing daily meteorological data. Authors are also thankful to the developers of satellite products, which helped us to perform different analyses in the manuscript.

**Conflicts of Interest:** The authors declare no conflict of interest.

## Appendix A

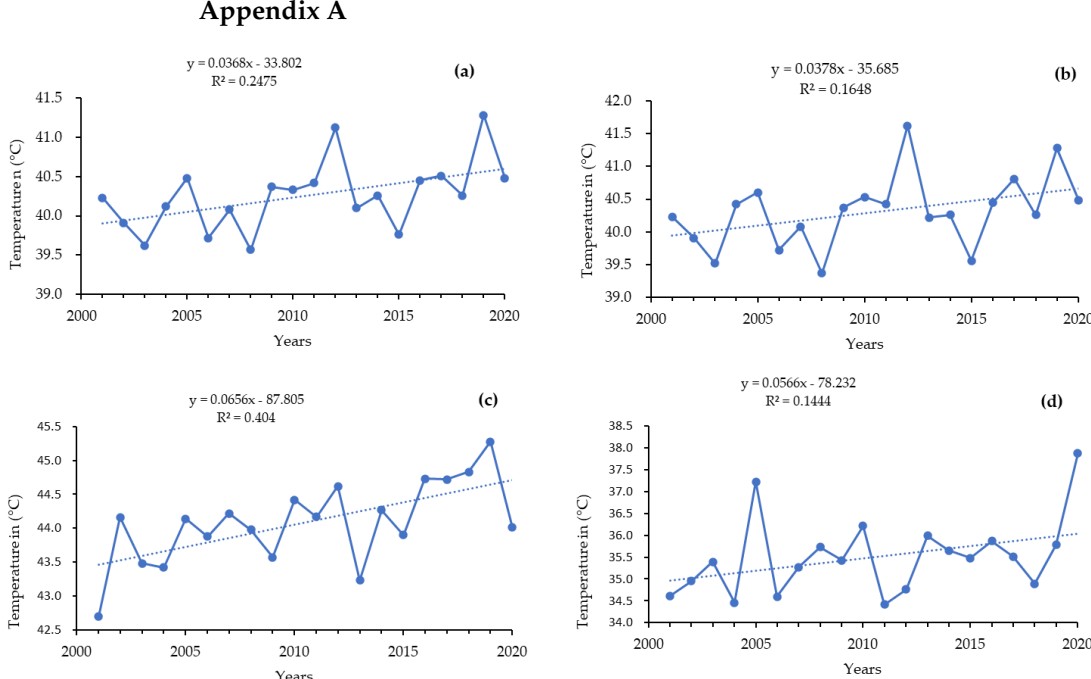

**Figure A1.** Variation trends in temperature from 2001–2020 at stations (**a**) TA 109, (**b**) TA 233, (**c**) J107, and (**d**) J108.

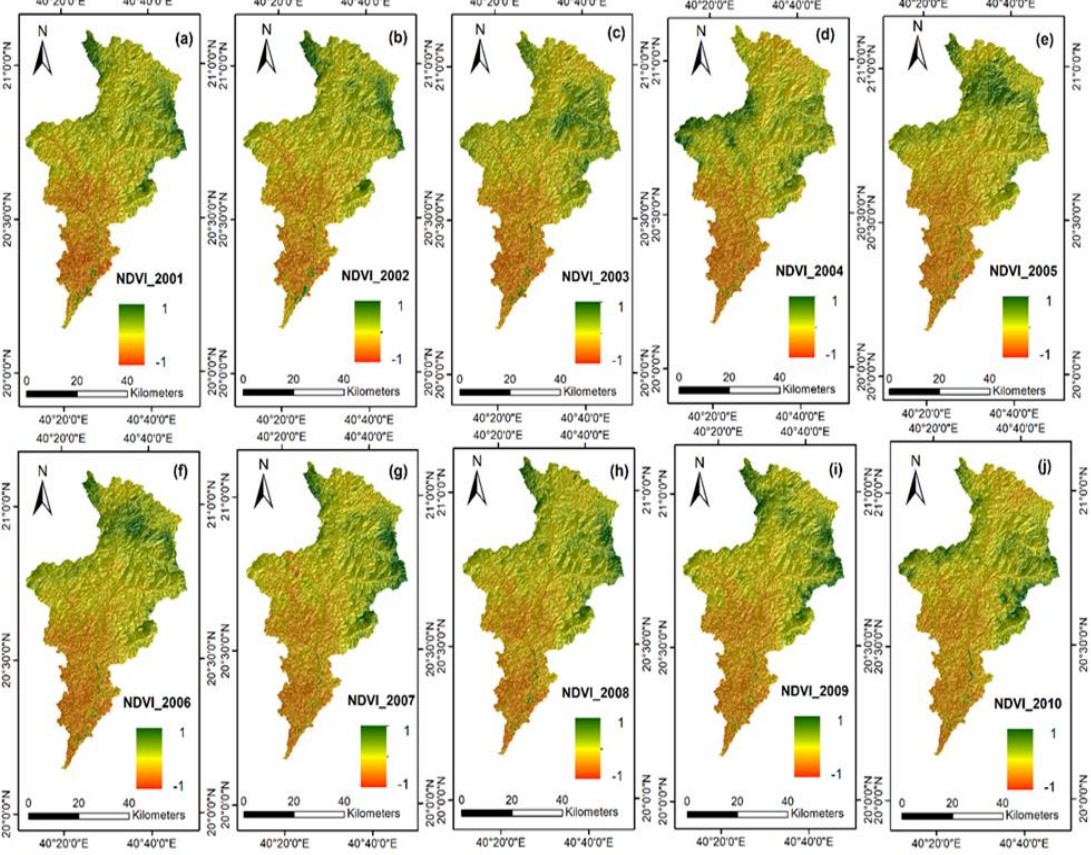

**Figure A2.** *Cont.*

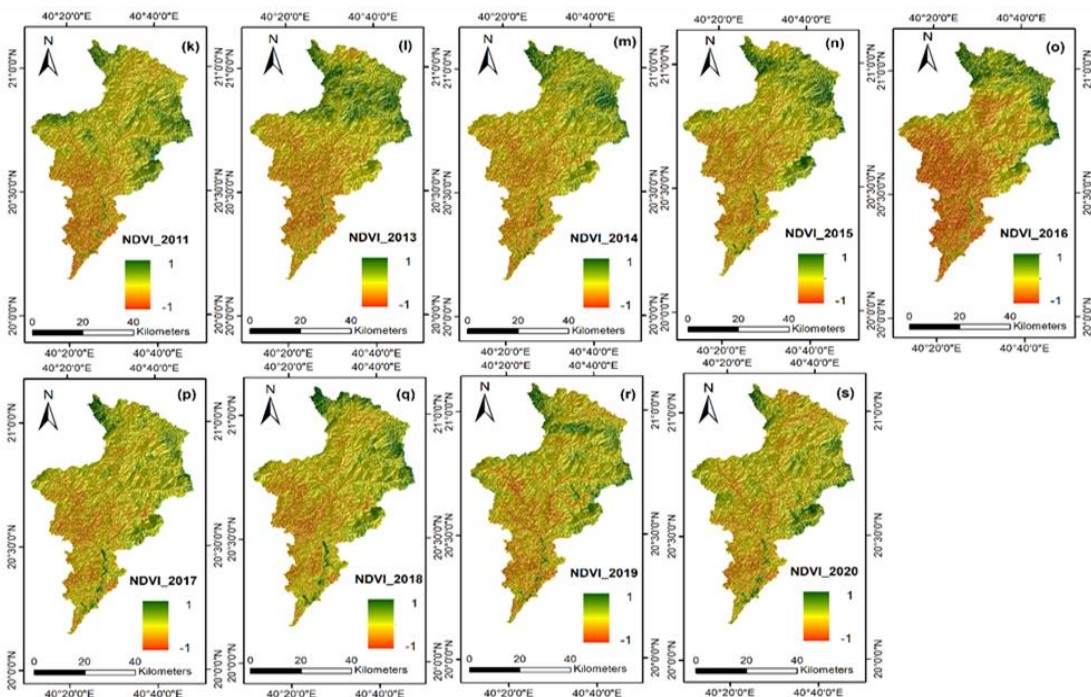

**Figure A2.** Spatial distribution of NDVI in Al-Lith Watershed retrieved from Landsat Satellites for the period of 2001–2020 (**a**–**s**).

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
