# Peer review of "Drought Monitoring Using Landsat Derived Indices and Google Earth Engine Platform: A Case Study from Al-Lith Watershed, Kingdom of Saudi Arabia"

_remotesensing, doi:10.3390/rs15040984_

Round 1

Reviewer 1 Report

This is interesting and meaningful study, but it needs significant improvement, so it needs major revisions. Please read my comments below.

1. It is not obvious how the GEE is applied in this study. Please provide more information.

2. Line 121-123, please rewrite them. The maximum monthly precipitation is 21 mm for all three months?

3. Line 154. What does limited data requirement mean? Please rewrite the sentence.

There are a lot of equations in this manuscript. Please ensure that all variables and the units of the variables have clear explanations.

For example, equation 1, what's the unit of T, Tmin, and Tmax? Kt is an empirical coefficient. What is its value.

Equation 2, what does i indicate?

5. In section 4.1.1 SPEI, the authors shows SPEI results on different time scales.

SPEI is calculated for the entire study region? Is it appropriate to do so?

What is the definition of a different time scale? 1-, 3- 6-, and 12-months indicate 1-, 3-, 6-, and 12-months running mean? Please make it clear in the manuscript.

If different time scales of SPEI are calculated by running mean or other methods, how could they use the same threshold for different extreme events? In other words, I don't think the values in Table 2 are applicable to all time ranges.

If the author uses the same threshold for different time scale, it is obvious SPEI-1 has more extreme events than SPEI-12, since the extreme events are smoothed out when doing the running mean, and it is weird that sometimes, the SPEI-3 has more extreme values than SPEI-1. If the results is correct, then what's the explanation?

What's the purpose of computing SPEI at different time scales?

Figure 3 is really hard to read. Please try just using different colors to indicate different time scale and using horizontal lines to represent the thresholds. Also, add tick marks on the x-axis.

6. When the authors show the results, please add more explanations rather than just describing them. In 2001, for example, the spatial distribution of VCI (Figure 4 a) is uniformly dry, while that of TCI (Figure 6 a) is a dipole. Is there any explanation for this?

7. In table 5, why is the correlation between VHI/TCI/VCI and SPEI usually increasing with longer time scale? Are VHI, TCI, and VCI monthly data?

8. What are the advantage of RSDI? To provide spatial information? I see some comparison results between VCI and TCI, so you suggest that VHI, the average of VCI and TCI, is a better index?

Reviewer 2 Report

Some questions are followed as:

 1.     Line 166-168: What is the “WB”? How did you get the values of parameters in Equation (4)?Must they be these constants or others?

2.     Line 207 & Equation (12): Does the LST equal to the Ts? The retrieved LST products might be composite temperature mingled crop canopy temperature and soil temperature.

3.     According to the Equation (14) and the proposed value of a = 0.5, the index of VHI was not an independent variable, just a half value of VCI and TCI. So, how does it make sense to analysis in this study?

4.     There was no Landsat Satellites images in 2012 based on the presentation in Table 1. How did you finish the results of 2012 in Fig. 3, 5, 7, 9, and table 4?

Round 2

Reviewer 1 Report

The authors have addressed all my questions. I think the paper is ready for publish.

Reviewer 3 Report

the manuscript is now acceptable.